

# Morphological variation of the relictual alveolar structures in the mandibles of baleen whales

Carlos Mauricio Peredo[1,2,3]    Nicholas D. Pyenson[1,4]

[1] Paleobiology, National Museum of Natural History, Smithsonian Institution, Washington D.C.,
United States of America
[2] Earth and Environmental Science, University of Michigan - Ann Arbor, Ann Arbor, MI,
United States of America
[3] Marine Biology, Texas A&M University - Galveston, Galveston, TX,
United States of America
[4] Paleontology and Geology, Burke Museum of Natural History and Culture, Seattle, WA,
United States of America

## ABSTRACT

Living baleen whales (mysticetes) are bulk filter feeders that use keratinous baleen plates to filter food from prey laden water. Extant mysticetes are born entirely edentulous, though they possess tooth buds early in ontogeny, a trait inherited from toothed ancestors. The mandibles of extant baleen whales have neither teeth nor baleen; teeth are resorbed *in utero* and baleen grows only on the palate. The mandibles of extant baleen whales also preserve a series of foramina and associated sulci that collectively form an elongated trough, called the alveolar groove. Despite this name, it remains unclear if the alveolar groove of edentulous mysticetes and the dental structures of toothed mammals are homologous. Here, we describe and quantify the anatomical diversity of these structures across extant mysticetes and compare their variable morphologies across living taxonomic groups (i.e., Balaenidae, Neobalaenidae, Eschrichtiidae, and Balaenopteridae). Although we found broad variability across taxonomic groups for the alveolar groove length, occupying approximately 60–80 percent of the mandible's total curvilinear length (CLL) across all taxa, the relictual alveolar foramen showed distinct patterns, ranging between 15–25% CLL in balaenids, while ranging between 3–12% CLL in balaenopterids. This variability and the morphological patterning along the body of the mandible is consistent with the hypothesis that the foramina underlying the alveolar groove reflect relictual alveoli. These findings also lay the groundwork for future histological studies to examine the contents of these foramina and clarify their potential role in the feeding process.

## INTRODUCTION

The diversity and ecological success of baleen whales (mysticetes) has been linked to dramatic evolutionary transformations in their feeding mechanisms (*Pyenson, 2017*; *Slater, Goldbogen & Pyenson, 2017*). Living baleen whales depart from the macroraptorial feeding of their toothed ancestors and instead bulk filter feed using keratinous baleen plates

Corresponding author
Carlos Mauricio Peredo,
cmperedo@umich.edu

(*Marshall & Goldbogen, 2016*; *Marshall & Pyenson, 2019*). Baleen whales are born entirely edentulous, although fossil relatives have mineralized teeth as adults. Embryological evidence demonstrates extant mysticetes briefly develop teeth *in utero* before resorbing them prior to birth (*Lanzetti, 2019*; *Lanzetti, Berta & Ekdale, 2018*; *Peredo, Pyenson & Boersma, 2017a*).

Because teeth are resorbed *in utero* and baleen develops only on the palate, the body of the mandibles of extant baleen whales lack any specialized feeding structure (*Peredo et al., 2017b*). Nonetheless, the mandible is still essential to the feeding process (*Goldbogen et al., 2017*; *Pyenson, Goldbogen & Shadwick, 2013*; *Shadwick et al., 2017*) and at least some mysticetes have evolved novel sensory organs that facilitate feeding (*Ford Jr, Werth & George, 2013*; *Pyenson et al., 2012*). Understanding how the edentulous mandible facilitates mysticete feeding is crucial to understanding the ecological transitions associated with the evolution of bulk filter feeding.

The mandibles of baleen whales preserve a series of dorsal foramina and associated sulci, often in such close proximity that they form a shallow groove (Fig. 1). These foramina have long been considered vestigial homologues to the dental alveoli of the resorbed teeth and the shallow groove has been termed the "alveolar groove" or "alveolar gutter". *Mead & Fordyce* (*2009*; page 42) consider "alveolar groove" to be the preferred terminology and review several other synonymous names for the structure. Recently, some authors have questioned this homology, and instead suggested that they represent distinct branches of the inferior alveolar artery or nerve and thus may have a vascular or nervous function (*Peredo et al., 2017b*). However, other authors have identified similar foramina in other edentulous mammals (*Ferreira-Cardoso, Delsuc & Hautier, 2019*), indicating that they may indeed be homologous to alveolar structures, though leaving it uncertain if these structures are vestigial or have some function.

Understanding the homology and potential function of these structures has been hampered by a lack studies detailing their morphology and variability. Here, we describe and quantify the observed morphology of these structures across the diversity of extant mysticete taxa. We report the number of foramina, the lengths of the alveolar region, and the lengths of the relictual alveolar foramen as defined by *Pyenson et al. (2012)* (the expanded distalmost foramen) for representative taxa of all four extant clades (including $n = 34$ specimens) and compare these values across taxonomic groups. Our results provide the anatomical context necessary for subsequent histological studies to examine the contents of these foramina and thereby elucidate their potential role in the feeding process.

## MATERIAL AND METHODS

We examined the mandibles of 34 baleen whales deposited in the collections of the Department of Vertebrate Zoology's Division of Mammals at the Smithsonian Institution's National Museum of Natural History. Our dataset includes members from all four major taxonomic groups. For each specimen, we report six anatomical measurements from one mandible of a pair (Fig. 1): the straight length of the mandible (SL); the curvilinear length of the mandible (CLL); the number of mental foramina (#MF); the number of alveolar

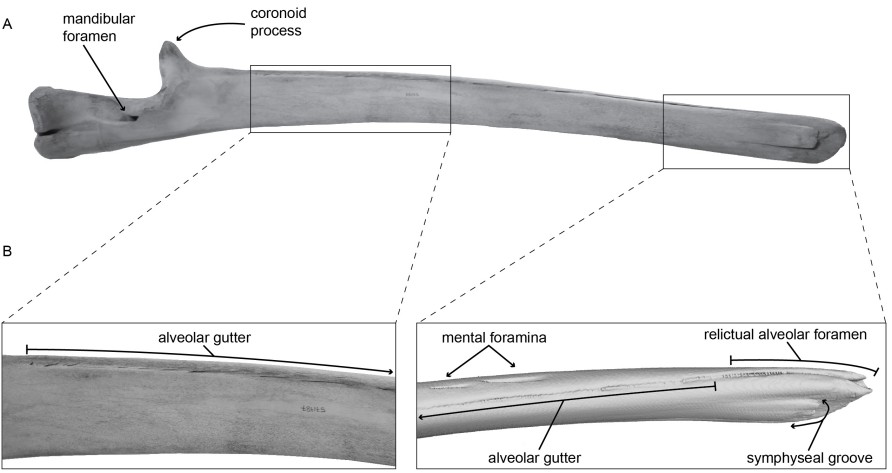

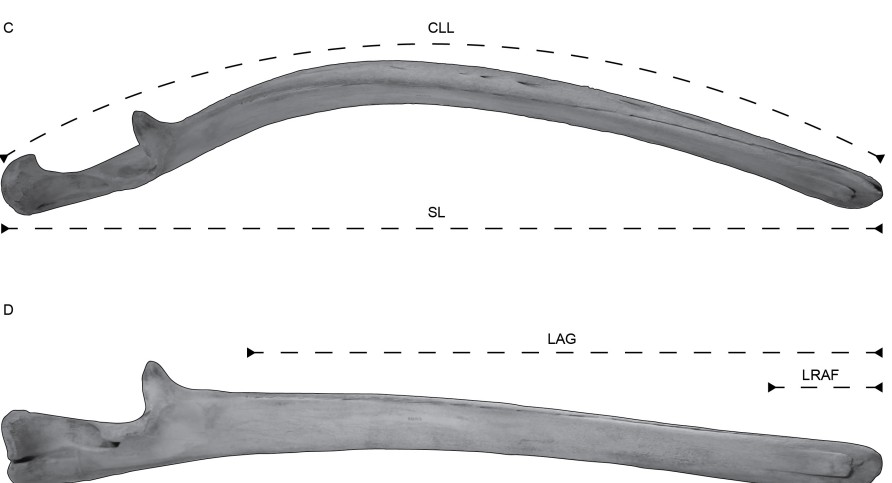

**Figure 1** **(A) Left mandible of *Balaenoptera acutorostrata* (USNM VZ 571487) in medial view with selected regions enhanced in panel (B).** (B) Enhanced view of the selected regions on panel (A), specifically highlighting the alveolar groove, associated sulci, the relictual alveolar foramen at the distal terminus of the mandible, and the symphyseal groove, using CT data. (C) Left mandible of *Balaenoptera acutorostrata* (USNM VZ 571487) in dorsal view demonstrating the curvilinear length (CLL) and the straight length (SL) measured in this study. (D) Left mandible of *Balaenoptera acutorostrata* (USNM VZ 571487) in medial view demonstrating the length of the alveolar groove (LAG) and the length of the relictual alveolar foramen (LRAF) measured in this study.

openings in the alveolar groove (#AL); the length of the alveolar groove (LAG); and the length of the relictual alveolar foramen (LRAF). We also report the length of the alveolar groove and the length of the relictual alveolar foramen as percentages of the total curvilinear length (Table S1). For smaller specimens, we took measurements using an anthropometer, and for larger specimens we took measurements using a measuring tape directly on the mandibles. The measurements for straight length (SL) and curvilinear length (CLL) follow
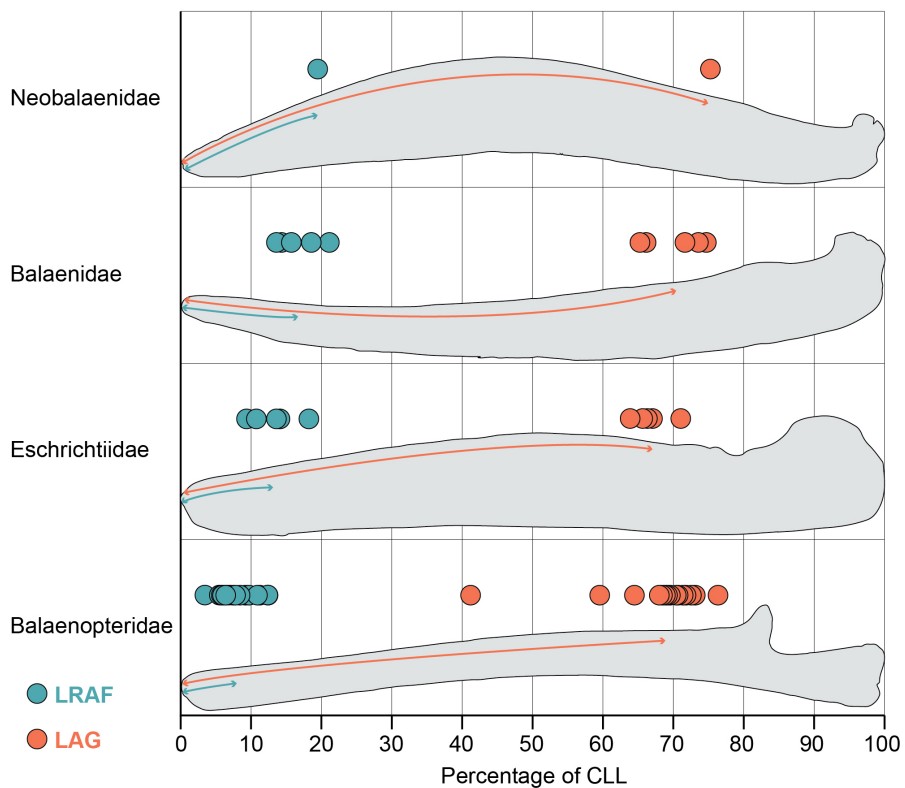

**Figure 2** **Length of the alveolar groove (orange) and the length of the relictual alveolar foramen (blue) as proportions of the total curvilinear length of the mandible.** This graph includes all adult specimens in our study but excludes the subadult and fetal specimens reported in Table S1.

(*Pyenson, Goldbogen & Shadwick, 2013*, Fig. 2). Additionally, we categorized each specimen based on its ontogenetic age: fetal, subadult, or adult based on suture fusion of associated cranial material (*Bisconti, 2001*; *Walsh & Berta, 2011*). Specimens were designated as fetal based on collection records and as subadult based on the degree of suture fusing in the skull. The final dataset includes four fetal, three subadult, and 27 adult specimens spanning 11 total species.

## RESULTS

Our dataset comprises 34 baleen whale mandibles including five balaenids, one neobalaenid, five eschrichtiids, and 23 balaenopterids, seven of which are fetal or subadult specimens. For the entire mysticete dataset, the number of mental foramina ranged from four (*Balaenoptera acutorostrata*) to nine (*Balaenoptera physalus*). The number of alveolar openings in the alveolar groove ranged from 10 (*Eubalaena australis* and *Eubalaena glacialis*) to 26 (*Balaenoptera physalus*). Overall, both the number of mental foramina and the number of alveolar openings in the alveolar groove varied within a species and neither showed clear taxonomic patterns.

The mandibles in our dataset span the full range of mysticete body sizes, resulting in a wide variation of mandible length. The smallest adult mandible (*Caperea marginata*) in our dataset is 126 cm in curvilinear length (CLL) and the largest has a CLL of 723 cm (*Balaenoptera musculus*). Consequently, the length of the alveolar groove varies widely across out dataset as well: the minimum recorded value of the alveolar groove in an adult mandible is 86 cm (*Eschrichtius robustus*) and the maximum recorded value is 513 cm (*Balaenoptera physalus*). However, this variation is relatively constrained proportional to the mandible's CLL (Fig. 2). Our results demonstrate that most taxa have alveolar groove lengths that are approximately 60–80 percent of the mandible's total CLL. This pattern was true across all four taxonomic families and in the fetal and subadult specimens as well. Only one whale in our dataset fell notably outside these values: USNM 571340 (*Balaenoptera borealis*) has an alveolar groove that is only 41% of the mandible's CLL. This anomalous datapoint may be the result of an unobserved pathology, ontogenetic variation, or linked to the unusual feeding biomechanics of sei whales (*Segre et al., 2021*).

Despite no taxonomic pattern in the proportional length of the alveolar groove, the proportional length of the relictual alveolar foramen (RAF) does vary by taxonomy (Fig. 2). Balaenids and the lone neobalaenid in our study have an RAF that is approximately 15–25% of the total CLL. However, most adult balaenopterids have an RAF that is only approximately 3–12% of the CLL. The proportional length of the RAF in balaenids is on average twice as long, and in some cases as much as five times as long, as the values observed in balaenopterids (Fig. 2). Eschrichtiid mandibles have values roughly between balaenids and balaenopterids, approximately 9–16% of the mandible's CLL.

This pattern does not, however, extend to the fetal and subadult samples in our study, all of which are balaenopterids. Interestingly, the subadult balaenopterids exhibit higher values for the proportional length of the RAF (13–17% of the CLL) and the fetal specimens preserve even higher values still (18–28% of the CLL). A full comparison of the allometry of these structures across whale ontogeny is beyond the scope of this project. However, our data indicate that fetal balaenopterids have RAF of similar proportional length to those of balaenids and suggest that the restriction of the RAF to the distal tip of the mandible may occur later in ontogeny.

Importantly, the variation in the proportional length of the RAF is not a function of overall length of the mandible. The largest (*Balaenoptera musculus*) and smallest (*Balaenoptera acutorostrata*) balaenopterids in this study both preserve some of the proportionately shortest RAF (approximately 5–6 and 3–9% of the CLL respectively). In contrast, *Caperea marginata*, which is comparable in size to *Balaenoptera acutorostrata*, has an RAF that is nearly 20% of the CLL, and the largest balaenids (*Balaena mysticetus* and *Eubalaena glacialis*) have RAF that are approximately 14 and 18% of the CLL. This suggests that, although the RAF becomes proportionately shorter throughout ontogeny in balaenopterids, the pattern is not being driven simply by ontogenetic growth to larger body size.

## DISCUSSION

The high degree of variability in the number of alveolar foramina present is noteworthy given that it remains unclear if the alveolar foramina and the alveolar groove are actually vestigial remnants of the dentition resorbed *in utero*. *Peredo et al. (2017b)* considered their homology with teeth uncertain, in part because the mandibles of baleen whales exhibit evidence of bone remodeling similar to the patterns observed during pathological tooth loss, where the alveoli are entirely resorbed, and the bony surface becomes solid. However, recent authors (*Ferreira-Cardoso, Delsuc & Hautier, 2019*) have identified similar structures in other edentulous mammals and considered them vestigial remnants of the resorbed dentition. Our findings suggest that the morphology of the alveolar groove and the relictual alveolar foramen are constrained by developmental pathways early in ontogeny, supporting the hypothesis that they are homologous with the resorbed dentition (see also below about anatomical patterning). If this is the case, then it is noteworthy that these foramina and internal canals are not resorbed during the bone remodeling process (*Peredo et al., 2017b*), and their morphological patterning remains consistent across all four extant families. The consistency of this pattern across extant mysticetes suggests that these foramina may have been co-opted to perform a novel function in specific lineages, as documented in the case of the chin sensory organ in rorquals (*Pyenson et al., 2012*). However, the chin sensory organ is notably absent in balaenids, suggesting that these foramina function may more simply be related to simply innervate soft connective labial eminences (see references in *Peredo et al., 2018*).

Moreover, if each alveolar foramen is the vestigial remnant of an individual alveolus resorbed *in utero*, then we would predict the high degree of variability in the number of alveoli present (ranging from 10 to 24) that we observed. Unfortunately, dental counts for embryonic mysticetes are rare, making it difficult to test this prediction. Recent work by *Lanzetti (2019)* and *Thewissen et al. (2017)*, which builds on that of *Ishikawa & Amasaki (1995)* and *Ishikawa et al. (1999)*, expands the histological datasets of early ontogenetic variability in mysticete dentition, especially across a taxonomic breadth that may elucidate an evolutionary framework to test the relationship between alveoli and tooth identity. Additionally, many 19th and early 20th century anatomists report tooth counts for foetal mysticetes, though this work is scattered across many languages and difficult to verify (see *Peredo, Pyenson & Boersma (2017a)* for a review of embryological and histological data pertaining to mysticete tooth buds).

Traditionally, the ramus is defined as the vertical, non-tooth bearing portion of the mandible (*Mead & Fordyce, 2009* and references therein). However, extant mysticete mandibles are single elongated osteological elements that lack an obvious distinction between the body of the mandible and the ramus. Our results demonstrate that proportional length of the alveolar groove is tightly constrained around an average of 70% of the curvilinear length of the mandible. This suggests that although extant mysticete mandibles have no obvious distinction between the body and the ramus, they retain the distinct anatomical patterning of their terrestrial ancestors. Based on the fossil record of mysticetes, this loss of a major distinction between the body and the ramus happened no later than

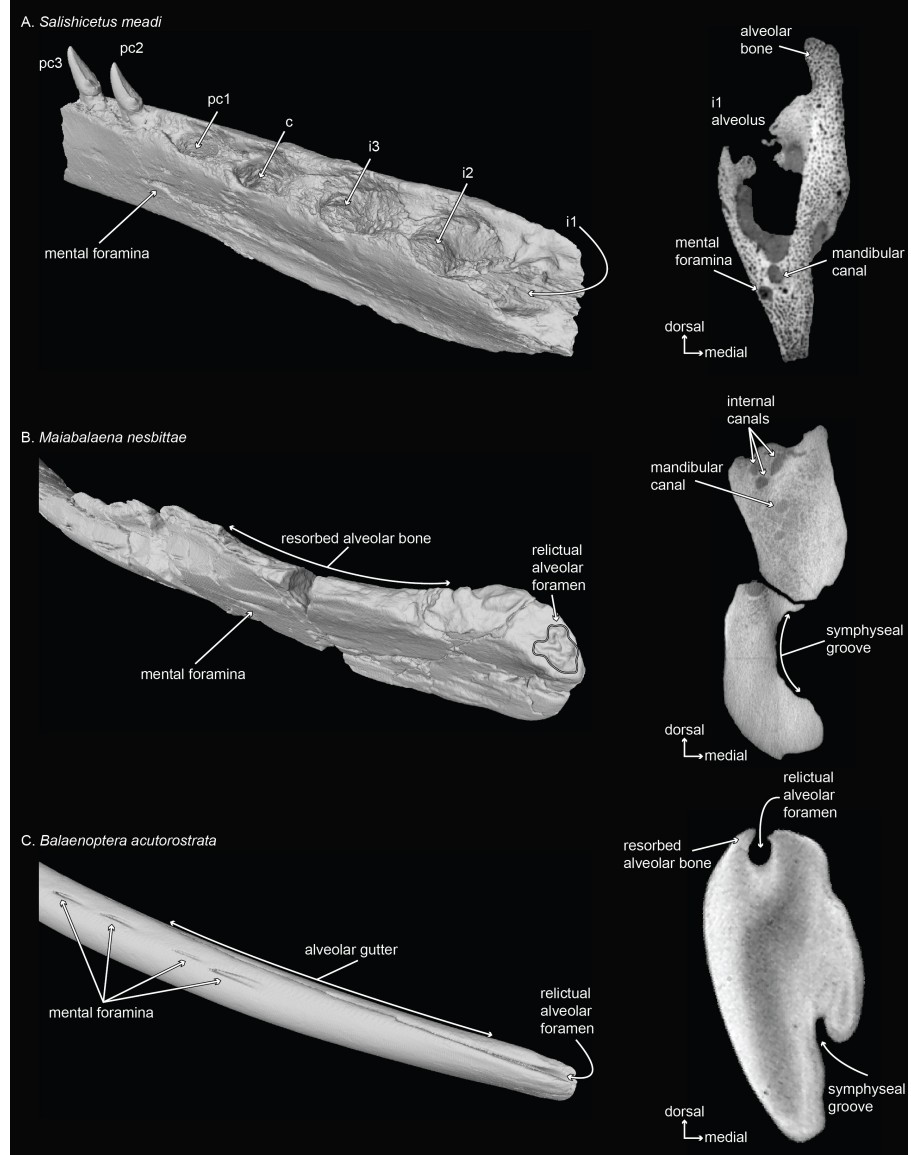

**Figure 3** **Mandibles of a toothed stem mysticete (A) UWBM 50004:** *Salishicetus meadi*; **an edentulous stem mysticete (B) USNM PAL 314627:** *Maiabalaena nesbittae*; **and an extant mysticete (C) USNM VZ 571847:** *Balaenoptera acutorostrata.* Mandibles are figured in oblique view and in cross sectional slices from CT scans to demonstrate the morphological variability of the distal alveolus or associated foramina.

the early Oligocene (*Peredo et al., 2018*), after the origin of aetiocetids but prior to the lineage leading to *Maiabalaena+ Sitsqwayk*, eomysticetids, and crown mysticetes (*Peredo & Pyenson, 2018*).

The co-option of the relictual alveolar foramen for novel functions in certain extant lineages (e.g., *Pyenson et al., 2012*) appears to be part of a longer trend in bone remodeling that has occurred in stem lineages leading to crown Mysticeti (Fig. 3; and see *Peredo et al., 2017b*). Although stem mysticetes with fully mineralized adult teeth (e.g., *Salishicetus*,

Fig. 3A) display no unusual patterns in this regard, the stem mysticete *Maiabalaena* shows extensive cortical remodeling in the dorsal margin of the mandible that is homologous with the alveolar groove in extant mysticetes (Figs. 3B, 3C). Interestingly, the complexity of the relictual alveolar foramen morphology in *Maiabalaena* is a trait that appears in crownward eomysticetids, such as *Waharoa ruwhenua*, which may have possessed mineralized teeth (*Boessenecker & Fordyce, 2015*). The diversity of relictual alveolar foramina in these stem lineages merits deeper examination to understand the range of alveolar morphology, any associated dentition, and the distribution of internal structures related to the mandibular canal and mental foramina (*Peredo et al., 2017b*).

Despite no taxonomic differences in the proportional length of the alveolar region for extant mysticetes, we report substantial taxonomic differences in the proportional length of the relictual alveolar foramen (Fig. 2). This foramen is the distalmost opening on the dorsal margin of the mandible and is elongated to as much as 20% of the CLL in balaenids and neobalaenids but constrained to only about 5% of the CLL in balaenopterids. Previous authors have reported that in balaenopterids, the relictual alveolar foramen is the opening through which the lunge feeding sensory organ in the chin is innervated (*Pyenson et al., 2012*). This sensory organ facilitates lunge feeding in balaenopterids and is absent in balaenids, but it remains unclear if a homologous structure is present in other mysticete groups. If the relictual alveolar foramen has been coopted to innervate the chin sensory organ, then this may constrain the length of the relictual alveolar foramen in balaenopterids.

**Institutional Abbreviations**

| | |
|---|---|
| **USNM PAL and VZ** | Departments of Paleobiology and Vertebrate Zoology (Division of Mammals), National Museum of Natural History, Smithsonian Institution, Washington DC, USA |
| **UWBM** | Burke Museum of Natural History and Culture, University of Washington, Seattle, WA, USA. |

# ACKNOWLEDGEMENTS

We would like to thank DJ Bohaska, AJ Millhouse, D Lunde, and JO Ososky for facilitating access to specimens at the NMNH. We further thank J Villari and C Robinson for assistance with data collection. Finally, we thank SB Sholts for access to the SIBIR CT facility and the Mimics Innovation Suite license.

## Funding

Carlos M. Peredo and Nicholas D. Pyenson were supported by the Remington Kellogg Fund and the Basis Foundation. Carlos M. Peredo was further supported by the National Science Foundation (NSF Award #1906181) and the Michigan Society of Fellows. The funders had no role in study design, data collection and analysis, decision to publish, or preparation of the manuscript.

## Grant Disclosures

The following grant information was disclosed by the authors:
Remington Kellogg Fund and the Basis Foundation.
National Science Foundation (NSF Award: 1906181.
Michigan Society of Fellows.

## Competing Interests

Nicholas D. Pyenson is an Academic Editor for PeerJ.

## Author Contributions

- Carlos Mauricio Peredo and Nicholas D. Pyenson conceived and designed the experiments, performed the experiments, analyzed the data, prepared figures and/or tables, authored or reviewed drafts of the paper, and approved the final draft.

## Data Availability

The full raw data is available in the Supplementary Files.

## Supplemental Information

Supplemental information for this article can be found online at http://dx.doi.org/10.7717/peerj.11890#supplemental-information.

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
