# Peer review of "Morphological variation of the relictual alveolar structures in the mandibles of baleen whales"

_PeerJ, doi:10.7717/peerj.11890_

## Round 0.1 · original submission · Minor Revisions

Dear authors,

Thank you for your submission to PeerJ. After moderate revisions, I think that this manuscript will make an excellent contribution to the literature. Please pay careful attention to reviewer comments and submit a clean version of your manuscript, a tracked changes version, and an itemized response to reviewers document with your revision.

In particular, reviewer 1 noted that you should add more background and context to the study and more explanation of how measurements were taken in the methods. Reviewer 2 notes areas of the manuscript that need to be expanded upon with more references and additional discussion to justify the support for your predictions. Specifically, you should limit conclusions around the Neobalaenidae given that they are based on a single taxon.

Please let me know if you have any questions regarding your revision. Thank you for your submission.

Best,

Brandon P. Hedrick, Ph.D.

·

Basic reporting

This is a short, sweet manuscript that nicely lays out the study, the historical context of the issue (although a bit more might be said on that front), and sufficiently describes its findings and their importance. Raw data are supplied and easy to examine.

Experimental design

The actual scientific study is simple and is for the most part adequately explained here, although some terms should be clarified better and simple methods might be outlined in more detail. However, I have no qualms about the methods--they seem to have been entirely appropriate and are mostly detailed well here.

Validity of the findings

The findings are clearly described in the text and shown in a figure, and their impact is discussed, although I suggest below a couple of other factors for the authors to consider.

Additional comments

Review of Peredo and Pyenson PeerJ 61530, “Morphological variation of the relictual alveolar structures in the mandibles of baleen whales”

The issue of the transformation of Mysticeti from early toothed mysticetes to entirely edentulous filter feeders, with only small, temporary tooth buds, is an important topic that has seen a resurgence of interest in recent years. This study addresses an important point: do the grooves, sulci, foramina, and other depressions on the mandibles of extant mysticetes relate to the presence of incipient tooth buds, and the former presence of erupted teeth, and if so how? Or is there no connection whatsoever? This is a topic that is of broad interest and one that deserves continued attention.

This manuscript takes steps toward addressing these questions, but it could go a bit further. One of my concerns is that the terminology itself might be clarified more with a bit more of a literature review. This manuscript suggests (line 50) that the terms “alveolar groove” and “alveolar gutter” are synonymous, but that isn’t certain. Who has used, or introduced, these terms?

More importantly, it is not clear exactly what the relictual alveolar foramen is, and I recommend the authors precisely define and describe what constitutes this foramen and what its relation is to the alveolar gutter. Based on Figure 1B it seems these are entirely separate and distinct features of the mandible, but based on Figure 2 and the text, these two features often overlap partially or entirely. Is it just “the expanded anteriormost foramen” (line 59)? If so, this seems to be somewhat murky and subjective. Is there an objective way to identify where the relictual alveolar foramen ends?

This leads to a related point regarding methods. This is at heart a simple study, which can be a very good thing, but it would be good to read a bit more about how measurements were taken, particularly for measures like the curvilinear length. Just by using tape measures on bones themselves, or by analyzing scaled photos with a computer program, etc.?

Overall, I appreciate the crisp brevity of this manuscript (concision is always to be applauded), and I don’t advocate making it much longer, but I do think there are a few factors the authors should consider (and potentially address), including both paleontological and embryological studies.

I welcome the conclusion that there are different patterns in the proportional length of the relictual alveolar foramen in different mysticete taxa, with the proportional length being considerably longer in balaenids than in balaenopterids. This raises the obvious question of whether anything is known about teeth in the ancestors of these families. Do we know of toothed ancestors either directly in these two families, or in earlier mysticetes allied with these two lineages? It would be nice to know how, if at all, the jaw patterns reported here align with previously reported findings on dental diversity—which would inform the analysis here, make it more useful, and put it into perspective for readers. That of course is presuming there is enough known to say anything on that point…

In addition to paleontological data, what (and how much) is known of developmental data about tooth bud counts in different mysticete taxa? If differences exist and are known, do they correlate with differing relictual foramen lengths, or with other variables (such as the number of baleen plates, for example)? I don’t know much about dental anlagen, but I do know various people have looked a fair bit at them, and I believe there is data both for bowhead/right whales as well as rorquals.

Why does Balaenoptera borealis fall “notably” outside the dataset relative to other whales (and other rorquals)? Even if the authors don’t know, they might speculate about this curious finding. Is there anything notably different about sei whales and their development, morphology, or feeding ecology?

I like the paragraph in the Discussion about the mandibular ramus vs. body.

The two figures are nicely constructed and easy to follow. They are of suitably good quality.

Raw data has been supplied and can also be followed easily. The analysis and findings seem valid.

The text is good, with clear writing. I have a few very minor suggestions, but these are really trifling.

Line 12: some would clarify that not all baleen feeding truly involves sieving; perhaps safer to say simply that baleen plates are always used to filter food from water
Line 13: “Today’s mysticetes” makes sense but seems oddly informal. Perhaps extant (even though that word is used in the next sentence) or living or modern?
Line 21: There seems to be a word missing. Broad variability across what?
Line 51: Recent authors have (not has)
Line 55: the sentence here is OK but I had to do a double take with the final “though leaving it uncertain” clause, which refers back to the other authors (not the alveolar structures). I’d rewrite this, but it can probably stand.
Line 78: why not clarify eleven total species (not just taxa)

Reviewer 2 ·

Basic reporting

Need to cite a recent paper Ekdale and Demere, 2021 that describes lower jaw anatomy of an extinct toothed mysticete, important for comparison with authors’ findings of similar structures on jaws of extant mysticetes.

Experimental design

The ms addresses an important research question that thus far has received little attention- description of the basic anatomy of the lower jaw of mysticetes with implications for the presence of teeth and their potential function in feeding. Strengths of the paper include detailed description of variability and quantification of morphology of lower jaw among representative mysticete taxa from all extant clades (families). An important variable ontogenetic age was considered with sampled mandibles from fetal, subadult and adult (the majority) specimens. Although the question of the homology of mandibular dorsal foramina to dental alveoli will need to be examined in histological study this descriptive paper lays the necessary anatomical foundation for that further work.

Validity of the findings

A few weaknesses were noted that should be addressed. Taxonomic sampling good with one exception. For one extant family sampled Neobalaenidae (with only 1 taxon) only a single specimen was examined which limits the reported results of variability in this family. I understand that there is only one specimen of this taxon in the US but others exists outside of the US thus presenting an opportunity for international collaboration and more meaningful results. The authors favor the conclusion (really a speculation without further study) that the alveolar gutter and relictual alveolar foramina (RAF) are homologous with resorbing dentition that have been co-opted to innervate the chin sensory organ or innervate labial eminences--both need more explanation vs the authors’ mention of appropriate references with no further discussion. If they think their results support this then they need more evidence. Importantly, balaenids (and neobalaenids?) lack this sensory chin organ but possess the foramina. Their data at this point consists entirely of a suggestion with limited evidence for one taxon of “constrained developmental pathways early in ontogeny.”

---

## Round 0.2 · accepted · Accept

Dear authors,

Thank you for your submission to PeerJ and for your corrections to a prior round of reviews. I am pleased to move this manuscript forward to the proof stage. I am excited to see it published!

Upon a final reading, I did notice a few grammatical issues that should be fixed prior to publication:

Line 78: I would think ‘ontogenetic stage’ rather than ‘ontogenetic age’. Age suggests numbers.

Line 96: Change ‘out’ to ‘our’

Line 103: Space needed between ‘CCL.This’

Line 137: delete extra space before comma

Line 144: ‘suggesting that the function of these foramina may..’

Fig. rather than figure when mentioned in text

Please let me know if you have any questions.

Best,

Brandon P. Hedrick, Ph.D.

Reviewer 2 ·

Basic reporting

The ms has been successfully revised based on reviewers comment and is now ready for publication.

Experimental design

No comment

Validity of the findings

No comment